# Determinants of Care Pathways for C-PTSD Patients in French Psychotrauma Centers: A Qualitative Study

**DOI:** 10.3390/ijerph20136278

**Published:** 2023-07-02

**Authors:** Germain Salome, Philippe Vignaud, Perrine Galia, Nathalie Prieto, Nicolas Chauliac

**Affiliations:** 1Centre Régional du Psychotraumatisme Auvergne-Rhône-Alpes, Hôpital Edouard Herriot, Hospices Civils de Lyon, F-69003 Lyon, France; 2Faculté de Médecine Lyon-Est, Université Claude Bernard Lyon 1, F-69008 Lyon, France; 3Research on Healthcare Performance (RESHAPE), INSERM U1290 & Université Claude Bernard Lyon 1, Domaine Rockefeller, F-69008 Lyon, France

**Keywords:** C-PTSD, complex post-traumatic stress disorder, psychiatry, ICD-11, care management

## Abstract

In 2018, the International Classification of Diseases (ICD-11) established a novel nosographic category within the stress-specific disorders known as complex post-traumatic stress disorder (C-PTSD). Characterized by distinctive clinical attributes and a limited response to conventional PTSD treatments, C-PTSD has prompted the reconsideration of care methods. Our study’s purpose was to explore the intricate factors shaping the care pathways for individuals suffering from C-PTSD. We used a grounded theorization technique involving professionals across a range of specialized French psychotraumatology institutions. The resulting comprehensive theoretical model offers valuable insights into the constitution mechanisms of these pathways, helping elucidate the varying care options. Interestingly, we found that differences in clinical perspectives were determined by the care provider’s viewpoint on clinical guidelines, screening tools, and treatment options, but also by structural and organizational factors. The distinctive dynamics and interrelationships identified in our research reveal potential areas of focus for incorporating C-PTSD care more effectively into specialized French trauma centers. This investigation offers a path toward improved understanding and management of C-PTSD, ultimately advancing patient outcomes.

## 1. Introduction

In the 2018 update of the International Classification of Diseases (ICD-11), a new diagnostic category emerged, termed complex post-traumatic stress disorder (C-PTSD). This diagnosis broadens the scope of disorders specifically associated with stress, delineating itself distinctly from post-traumatic stress disorder (PTSD) [1]. While PTSD is characterized by three key symptom types—re-experiencing, avoidance, and hypervigilance—following a highly threatening event or series of events, C-PTSD involves these features and additional symptoms from three domains of disturbance in self-organization (DSO): negative self-perception, emotional dysregulation, and interpersonal disturbance [2]. This diagnosis builds upon the symptom criteria for disorders of extreme stress not otherwise specified (DESNOS), a concept first introduced by Herman in 1992 [3] and later featured in the *Diagnostic and Statistical Manual of Mental Disorders*, 4th edition (DSM-IV) [4]. DESNOS was defined by disturbances such as alterations in the regulation of affects and impulses, attention or consciousness, self-perception, relations with others, and in systems of meaning and somatization. This diagnosis, which was for a long time the clinical reference for the spectrum of complex or repetitive trauma among mental health professionals and was then also known as C-PTSD, is no longer officially part of the standard nosography (ICD or DSM). However, many of its criteria and clinical dimensions have been incorporated into new nosographic entities related to the psychotrauma spectrum, such as the ICD’s C-PTSD and dissociative disorders and the DSM’s dissociative and preschool subtype of PTSD [5,6].

Although similar types of events may cause both PTSD and C-PTSD, the ICD points out that C-PTSD is typically the result of prolonged or repeated exposure to situations where escape is challenging or impossible. The diagnosis of C-PTSD, grounded in clinical observations and validated by recent population studies [7,8], holds considerable clinical relevance due to its estimated global prevalence of between 0.5% and 7.7%, extending up to 36% in certain psychiatric inpatient populations [9]. Given the high prevalence and the well-documented long-term adverse physical and mental health outcomes of repeated or extended exposure to traumatic events [10,11], C-PTSD emerges as a significant public health concern. The clinical validity of this diagnosis is further confirmed by recent symptom network analyses that identify a uniform cluster of symptoms in populations exposed to potentially traumatic events. These analyses distinguish C-PTSD from other prevalent disorders like depression, emotionally unstable personality disorder, anxiety, or PTSD [12,13]. The International Trauma Questionnaire (ITQ), a specially devised psychometric scale, further aids clinicians in differentiating C-PTSD from PTSD. Moreover, recent studies have indicated a less pronounced clinical improvement in C-PTSD trauma survivors when treated with therapies traditionally used for PTSD, underscoring the necessity for innovative care and management strategies for this newly recognized condition [14,15,16].

In response to the series of terrorist attacks that unfolded in France in 2015, government entities, associations, and healthcare professionals undertook the structuring of psychological trauma care, establishing regional psychotrauma centers (*Centres Régionaux du Psychotraumatisme*, CRPs). These centers bear the responsibility of receiving, assessing, guiding, or treating individuals who have experienced a traumatic event while simultaneously creating a regional care network. However, recent societal acknowledgment of common situations potentially leading to C-PTSD and complex trauma—such as the #metoo movement [17], increased domestic violence during the COVID-19 lockdown [18], and reports on the developmental impacts of chronic violence [19]—has sparked a shift in the demands on the CRPs and the profiles of individuals who access these facilities.

The present research delves into the ascending relevance of C-PTSD within dedicated psychotrauma institutions. It aims, through qualitative exploration, to formulate a general theory that maps out the dynamics shaping the treatment pathway for C-PTSD within French CRPs and how these centers perceive their role in this pathway.

## 2. Materials and Methods

### 2.1. Study Design

A qualitative study, inspired by grounded theory, was conducted due to the method’s reputation for generating concepts and hypotheses when exploring new or underexplored questions in the literature. The objective of this inductive approach was to elucidate the underlying phenomena involved in the formation of the care pathway for clients with C-PTSD and to develop a comprehensive theory encompassing these phenomena through an ongoing process of data collection and analysis.

### 2.2. Participants Recruitment

A theoretical sampling approach was employed to progressively recruit participants who could contribute valuable insights into the intermediate hypotheses generated during the study. All ten French CRPs (at the time of the study) were contacted via email to invite professionals to voluntarily participate. The responses received were organized by CRP, and a recruitment list of CRPs was established based on the highest number of interested participants and the geographical feasibility of meeting them.

The collected data, obtained anonymously with participants’ consent, included information regarding their professional role, age, and years of experience in the field of psychotrauma. The inclusion criteria encompassed professionals working in a CRP with a minimum tenure of six months; involved in reception, evaluation, orientation, or care roles within the center; and willing to participate in the study. Recruitment ceased upon reaching theoretical data saturation, where further participants no longer added significant new insights.

### 2.3. Data Collection

Data were collected through semi-structured interviews conducted in French (Table 1) combined, when possible, with an open discussion on undiagnosed clinical situations of PTSD, C-PTSD, and DESNOS.

The interviews were recorded using a digital voice recorder. Prior to each interview, participants were provided with information regarding their rights to withdraw, rectify, and maintain confidentiality and anonymity. Verbal consent to participate was obtained, and recorded consent was also acquired. The recorded audio files were then transcribed into verbatim transcriptions, which were further anonymized by removing any identifying information such as names, locations, and specific details.

To enhance the relevance of the information collected during the interviews, interim findings were analyzed and discussed with the co-authors prior to each subsequent meeting with a CRP. This iterative process allowed for the refinement of the interview framework and aided in guiding the recruitment process.

Recruitment of participants and data collection were concluded after the fifteenth interview. This decision was based on the analysis of the fourteenth and fifteenth interviews, which did not yield additional data that could contribute to a more comprehensive characterization of the categories or phenomena under investigation. This indicated that the theory being developed had reached a state of saturation [20].

### 2.4. Data Analysis

Consistent with the grounded theory approach, data collection, analysis, and coding (open, axial, and selective) were conducted concurrently rather than following a linear sequence. This approach enabled ongoing internal comparisons to be made with the collected material, facilitating the identification of similarities and divergences. Exploring these comparisons contributed to a deeper understanding of the underlying phenomena. By engaging in simultaneous data collection, analysis, and coding, the research process allowed for a dynamic and iterative exploration of the data, fostering the emergence of new insights and enhancing the understanding of the phenomena under investigation.

#### 2.4.1. Open Coding

The initial phase of analysis involved a line-by-line examination of the verbatim transcripts using NVivo version 1.6.2 (QSR International, Doncaster, Australia). All transcripts were independently coded twice by two authors. The results from the two authors’ coding were then compared to triangulate the data. This rigorous process allowed for the identification of 381 unique original codes.

In conjunction with conventional analytical techniques [21] such as experiential coding, theoretical questioning, thought exercises, internal comparisons of the material, and note-taking, these original codes formed the basis for the formation of six categories that conceptualized the phenomena under study and their respective characteristics. Throughout this analysis, the categories were continually grounded in the raw data, ensuring their relevance and validity.

The coding process, along with the resulting heuristic map, provided the necessary theoretical distance required for subsequent axial and selective coding. These subsequent coding stages build upon the initial analysis, allowing for a more focused and refined understanding of the data and the development of theoretical insights.

#### 2.4.2. Axial and Selective Coding

The initial coding phase resulted in the creation of numerous categories, which initially lacked clear boundaries and relationships. To address this, an axial coding stage was introduced, following the coding paradigm proposed by Anselm Strauss [22]. This stage allowed for the prioritization of categories and the identification of a central category that encompassed the phenomena under study. It also aided in clarifying the constitutive nature of the categories and their interconnections. Through axial coding, the central category “Perceptions of C-PTSD symptoms and needs” emerged, along with five peripheral categories: “Conceptions of C-PTSD”, “Use of tools”, “Care strategies”, “Health networks”, and “Caregiver–client encounters”.

Selective coding was then conducted using CANVA software (CANVA Pty Ltd., Sydney, Australia) to visualize and progressively integrate the categories and their relationships within a comprehensive model. This model was continuously refined throughout the study through the simultaneous collection of data and the three types of coding. Attention was given to negative cases or counter-examples during the coding processes, as they provided valuable insights into contextual factors and variations in the interactions between the developed categories.

The results of the coding stages, carried out by the lead author, were regularly examined, discussed, and interpreted in collaboration with the co-authors. Furthermore, the intermediate findings were shared with the participants for their feedback and input, fostering a collaborative and participatory approach in the research process.

### 2.5. Trustworthiness, Reliability, and Ethical Concerns

The scientific rigor and reliability of the present study were ensured by adhering to the methodological principles of grounded theory. To maintain the validity, quality, and transparency of the research and its findings, the COREQ quality and validity grid for qualitative studies [23] was employed throughout the design and execution of the study.

A comprehensive triangulation process was conducted for open coding, involving two authors independently analyzing and comparing the data. Theoretical sampling was employed to select participants with diverse professional roles, experiences, and practices, thereby enhancing the richness and depth of the data.

The research team consistently reviewed and refined the semi-structured interview guides to minimize the subjective interference of the principal investigator. Interim findings were shared with all participants, gathering their comments and perceptions on the observed phenomena and aligning the data with the realities of their professional practice. This iterative process aimed to increase the groundedness of the data and mitigate the influence of the researchers’ subjectivity on the theoretical model.

The study obtained validation from the Ethics Committee of the University Hospitals of Lyon and adhered to the regulatory requirements in France, including a declaration to the Commission Nationale de l’Informatique et des Libertés (CNIL), ensuring compliance with data protection and privacy regulations.

## 3. Results

### 3.1. Participants

A total of 15 participants from different professions (psychiatrist, psychologist, nurse, and secretary) were interviewed in five different CRPs (Table 2).

### 3.2. Phenomena and Logics Interacting in the Construction of the Care Pathway

The data collected and the application of grounded theory facilitated the development of a conceptual model that illustrates the dynamics influencing the construction of the care pathway for individuals with C-PTSD in CRPs. This model comprises a central category, “Perceptions of the symptoms and needs of C-PTSD individuals”, which is directly or indirectly connected to five peripheral categories: “Conceptions of C-PTSD”, “Use of tools”, “Care strategies”, “Health networks”, and “Caregiver–client encounters”. These categories and their interrelationships provide a framework for understanding the various factors and processes that shape the care pathway and contribute to a more comprehensive understanding of C-PTSD care within the CRP context.

#### 3.2.1. Conceptions of C-PTSD

During the interviews, a majority of the health professionals provided a definition of C-PTSD that closely aligned with the definition outlined in the ICD-11 classification system. Their understanding of C-PTSD was informed by their knowledge of the symptoms associated with disorders of extreme stress not otherwise specified (DESNOS), which are frequently linked to multiple or prolonged traumatic events of an interpersonal nature. About half of the professionals specifically defined C-PTSD according to the criteria outlined in ICD-11.

While there may have been some variations in the definitions provided, all the professionals agree that C-PTSD is readily identifiable and distinguishable from simple PTSD. Furthermore, they acknowledge that C-PTSD has a high prevalence rate.

The caregivers also agree that there are more addictive, mood, and anxiety comorbidities associated with C-PTSD symptomatology. Clinically, the similarity of symptoms between C-PTSD and complex trauma as defined by DESNOS increases the sensitivity of screening for C-PTSD, but may also lead to confusion in clinical assessment.

“… C-PTSD or complex trauma, I don’t feel competent to say that I know exactly what one is and what the other is. I wouldn’t be able to make a theoretical difference, but since the new ICD 11 definition, I have a compass to discuss with my colleagues in a common language”(Psychologist-I14)

There is agreement on the following three characteristics of C-PTSD:(i).The importance of the traumatic history in understanding the symptoms:

“In most cases, the multiplicity of traumatic events in the patient’s life course will allow us to predict whether the clinic will be associated with simple or complex.” (EMDR nurse practitioner-I2)

(ii).The functional impact of C-PTSD through interpersonal or emotional dysregulation, social and occupational repercussions, and changes in the client’s agentivity are key assessment parameters in the treatment strategy.(iii).Need longer, more complex and tailored care than other PTSD sufferers.

On the other hand, the caregivers hold varying perspectives regarding the potential presence of personality changes in individuals with C-PTSD. They encounter difficulties due to the close relationships between symptoms and a history of traumatic experiences that can exist for C-PTSD and personality disorders characterized by emotional instability or borderline patterns of behavior:

“I often ask myself: are we talking about a borderline personality or a traumatic personality in these people?”(Psychologist-I13)

These different perspectives lead to varying interpretations:-Certain professionals do not consider personality changes a specific focus for specialized intervention tasks in the field of psychotrauma care. Instead, they propose that these changes, viewed as a comorbidity that could be present in this population, necessitate general mental health interventions:

“… As far as complex traumas are concerned, related to abuse since childhood, where we sense that the young woman is in a borderline state and so on, I stick mainly to general psychiatric care and not to something that stigmatizes her in the question of trauma, avoiding “it’s all about the trauma”, when in fact there are many other things behind it and there have been changes in the personality, before or after the trauma, I don’t know… But often it goes far beyond the question of trauma and many other psychiatric issues are involved.”(Psychologist-I5)

-In contrast, other professionals believe that traditional psychological and psychiatric healthcare tends to overlook these aspects of personality change and disorders. They argue that these elements warrant the attention of CRPs, as recognizing the connection between trauma and personality opens up new ways of conceptualizing symptoms and providing comprehensive support to clients:

“… in fact, what people diagnosed as borderline personality, in 95% of cases, it is complex post-traumatic stress disorder, with a disturbance in attachment, interpersonal relationships, emotional security, and so on. (…) I find that the strength of trauma is that it gives them an angle to work on their personality… Once we explain that these are ultimately people who have experienced such and such a trauma at such and such a time in their lives that it has disturbed them so much that it leads to a presentation of their personality. It allows us to conceptualize things differently and to accept a different kind of care.”(Psychiatrist-I9)

As a result, the overlapping symptoms shared with emotionally unstable personality disorder (ICD-11) or borderline personality disorder (DSM-5), depending on the theoretical framework used, often lead to confusion during diagnostic assessments. The likelihood of individuals with C-PTSD being referred to general mental health services varies depending on the level of awareness and sensitivity of CRPs toward the impact of traumatic events on personality changes.

The positioning of professionals in relation to psychiatric and psychological references, as well as their adherence to specific theoretical frameworks, significantly impacts and contributes to the variability in conceptualizations of C-PTSD. Individual theoretical orientations and cultural influences further shape professionals’ perspectives and understanding of C-PTSD:

“… I think that, as in all disciplines, we have lenses and that certain approaches mean that we do not hear, listen, welcome and accompany or consider care in the same way, whether we rely on this or that clinical hypothesis.”(Psychologist-I14)

Indeed, there appears to be a “center effect” where dominant theoretical cultures influence the emphasis placed on different concepts related to C-PTSD. Depending on the specific center, various concepts such as dissociation, attachment theories, cognitive and behavioral patterns, developmental trauma, psychodynamic approaches, and more may be prioritized.

The selection of a specific nosological framework is closely intertwined with the chosen care project, thereby influencing the clinical criteria and dimensions considered relevant for treatment. Consequently, this choice impacts the utilization of assessment and treatment tools. As the following quotation exemplifies, the adoption of a particular nosographic framework, including the concept of C-PTSD, is influenced by the type of care project selected by the center:

“If we really want to study the improvement of a symptomatology linked to a therapy on complex trauma with the ICD 11 criteria, the criteria are relevant. Then, on the clinical evaluation, here, we are really working on the essentially PTSD dimension part, whether it is simple PTSD or complex PTSD…so we will continue to use the DSM-5.”(Psychiatrist-I1)

#### 3.2.2. The Use of Diagnostic or Therapeutic Tools: A Central Issue in the Organization of Care

The utilization of diagnostic and therapeutic tools by CRPs plays a crucial role in the assessment and development of care pathways. Standardized psychometric scales, which have been validated in the scientific literature, are commonly employed by most CRPs. Examples of such scales include the Life Event Checklist for DSM-5 (LEC-5), PTSD Checklist for DSM-5 (PCL-5), and International Trauma Questionnaire (ITQ), among others. However, the purpose for which these scales are used may vary among professionals.

Some professionals employ these scales to thoroughly explore the psychotraumatic symptomatology by utilizing three or four different scales. Others, on the other hand, use them to determine whether brief therapeutic approaches, such as cognitive behavioral therapy (CBT), eye movement desensitization and reprocessing (EMDR), or lifespan integration (LI), are appropriate for the clinical situation. CRPs that prioritize the use of psychometric scales to identify indications for brief therapy tools often rely on only one or two scales (e.g., the PCL-5), which limits the number of potential treatment targets identified and, consequently, diminishes the therapeutic potential of brief therapy. This utilization of scales has been described as a way to regulate the level of care provided to each individual.

Given the lack of recommendations and published data regarding the treatment of C-PTSD, the chosen assessment strategy can lead to diverse clinical perceptions based on the purpose of the assessment. The proposed treatment primarily revolves around the use of brief therapy tools (such as CBT, EMDR, or ICV), occasionally supplemented by systemic or psychodynamic therapy. The adoption of these tools is influenced by the three ideological attitudes listed below:-Brief therapy is used to treat the core symptoms of PTSD (intrusion, hypervigilance, avoidance) and then externalized care for other dimensions, symptoms, or post-traumatic personality functioning.-Brief therapy is used when other treatments fail by CRPs that are skeptical about brief therapy because the treatment goals and their duration are considered too restrictive.-Brief therapies are employed to address both the core symptoms of PTSD and additional symptoms, although it is acknowledged that they may provide only partial improvement for all the symptoms present. The number of therapy sessions is limited, but the treatment objectives can extend beyond the basic symptoms of PTSD. This includes conducting a comprehensive assessment and offering support for overall functioning, personality, and relationship work. If the client desires further assistance, a referral can be made to a private practice psychologist with expertise in psychotrauma for continued treatment.

It appears that the use of assessment tools and specific psychotherapies, in synergy with the influence of theoretical references, leads to an opening or focusing of perceptions of clinical needs and treatment goals. The opening of clinical perceptions through in-depth exploration beyond its role as a brief therapy indicator makes it possible to improve the effectiveness of care in some CRPs. For instance, the implementation of psychoeducational groups addressing dissociation and emotional regulation not only facilitates but also enhances the accessibility and effectiveness of subsequent psychotherapy for trauma survivors. In essence, depending on the chosen theoretical framework and tools employed, this approach can broaden the detection of symptoms, leading to the integration of their treatment into the overall care pathway, whether it is conducted within the CRP or outsourced to external providers.

“There are people who have too many dissociative symptoms that they find difficult to control and stabilize. So, we have developed a group therapy for them…”(Psychiatrist-I9)

Additionally, it seems that the failure of psychotherapies and treatments leads to a search for deeper clinical exploration to understand and adapt assessment and care.

“… In the case of one woman who had started EMDR therapy, which was very trauma-focused, we did not adequately assess the impact it would have in the hours that followed. And how the consequences of that impact would be handled by the external environment. We then said to ourselves that we hadn’t assessed the situation sufficiently and hadn’t put up enough safety nets to contain it…”(Child psychiatrist-I11)

Several professionals emphasize the significance of cultivating a diverse and flexible theoretical foundation to ensure a comprehensive understanding and a wide range of treatment options for clients.

“I trained myself [in EMDR] because I realized that the question of language, […] the stakes of internal conflictuality were not enough to accompany what was happening directly in the person’s body and the way it was affected”.(Psychologist-I14)

#### 3.2.3. Perceptions of Care Needs

In addition to the clinical framework and individual professional perspectives, the perception of symptoms and care needs is also influenced by national and local dynamics. While all CRPs strive to adhere to national requirements and guidelines set forth by the National Resource and Resilience Center (CN2R), variations exist in their understanding of their mission scope as well as in the types and duration of therapies they offer. These differences can be attributed to various factors experienced by caregivers, including the following:-The socioeconomic and cultural diversity of the populations encountered, but also the reception capacities of the CRPs and the availability of downstream care professionals:

“There’s a reality on the ground that means we have to manage to adapt, and sometimes it’s not to say we have to circumvent, but sometimes the directives we have, we have to know how to minimize them…”(Psychologist-I13)

-Carers’ individual theoretical backgrounds and the “center effect” specific to each CRP influence their care plans, as this carer suggests:

“… I would alienate all my colleagues, because it’s not the… it’s the essence of the service, we come to this service to do this.”(Child psychologist-I6)

In addition, the perception of a specialist role for CRPs in the assessment and management of symptoms such as personality change, dissociative disorders, and emotional dysregulation varies and appears to be related to the capacity of the professionals outside the CRP:

“… Anyway, if it’s feasible on an outpatient basis, why not, but I find it would be difficult, given the strong demand for referral to the CRP”(Psychiatrist-I9)

This has led some CRPs to propose broad targets for their interventions, while others focus on a few targets to maximize their capacity to absorb new clients.

#### 3.2.4. The Encounter between Caregivers and Individuals, and Its Challenges

Many CRPs encourage direct contact from clients or their families, allowing for a brief collection of clinical information by the secretary. This initial step is typically followed by a telephone or face-to-face meeting with the CRP to clarify and assess the clinical situation. This organizational approach allows for early referrals in cases where the presenting clinical situation may not fit within the framework of the CRP, such as individuals without post-traumatic stress disorder (PTSD) or those who need treatment for a co-occurring disorder that needs to be addressed first. Some professionals have highlighted a drawback of telephone assessments, noting that individuals with C-PTSD may encounter difficulties in fully elaborating on their symptoms due to a lack of perceived security within the therapeutic relationship.

The remaining CRPs employ various professionals such as general practitioners, psychologists, and victim support associations as referral channels, thereby reducing the volume of referrals. This approach enables them to minimize waiting times, facilitate team discussions for each case, and ensure a comprehensive assessment before accepting individuals into the CRP or making appropriate referrals. However, this management strategy is also perceived as potentially limiting access to care:

“The disadvantage is that the person doesn’t necessarily want to talk to their doctor, or they don’t really have a doctor, and so they have to go to a doctor’s appointment first. So it can block some people.”(Psychologist-I13)

All the therapists who were interviewed emphasized the importance of psychoeducation and the concept of co-expertise between professionals and clients as fundamental to the treatment process. Consequently, professionals’ perceptions of symptoms and needs directly influence the information provided to clients, potentially leading to changes in how clients perceive their disorder and their expectations of care. By incorporating the subjective experiences of clients, clinical perceptions and the role of CRPs among professionals are also transformed. There is a consensus among professionals regarding the necessity of making the traditional framework of care more adaptable to accommodate the unique profiles of trauma survivors. This flexibility is deemed essential in order to establish a robust therapeutic alliance capable of withstanding the challenges of avoidance and ruptures that commonly arise within therapeutic contexts.

“…It is better to spend time adapting and building a very good relationship with the person. There will be less avoidance afterwards, rather than trying to deal with the trauma very quickly and not adapting, and the person will blow you off and not come back.”(Psychologist-I13)

#### 3.2.5. The Healthcare Network

The individual’s financial capacity plays a crucial role in shaping the care pathway and determining the accessibility of healthcare services beyond the CRP. This, in turn, affects the potential for outsourcing care that is provided free of charge within the CRP. One of the CRP’s missions is to develop a healthcare network to treat people outside its care to meet the growing need for PTSD/C-PTSD treatment. This role is directly influenced by carers’ perceptions of the symptoms and needs of people diagnosed with C-PTSD. Through their role as local referral and training support, their conceptions of C-PTSD influence the representations, diagnostic skills, and attitudes of regional mental health stakeholders and other key social actors in public or private practice:

“… We have opened up quite widely so that we can train professionals who specialize in early childhood as well as in the medico-social field. We also train lawyers, the police, school health workers… and then, of course, all health professionals.”(Child psychiatrist-I11)

Some CRPs also work to identify barriers to accessing other local victim support services. They organize drop-in sessions on their premises, using the trust placed in the center to create a safe first point of contact for victims with other key actors in the resilience-building process. These actors may include police officers, lawyers, or judges, who can provide information on victims’ rights and the legal process.

#### 3.2.6. Care Strategy

For certain CRPs, the core symptoms of PTSD remain the primary focus of treatment, while others prioritize addressing multiple dimensions within a specified timeframe, with a focus on functional impact. Irrespective of the reference framework and assessment tools employed, all CRPs emphasize that the quality and adaptability of care depend on finding a balance between individual care needs and the broader population-level considerations:

“To treat complex PTSD from beginning to end, i.e., to follow it in all its dimensions, requires several years of care. We have chosen not to be involved in this function, but only in the PTSD part, in order to be able to provide short-term care and, above all, to ensure a turnover and to be able to respond to requests for care. The demand for care is enormous and if we do not have a CRP with an absolutely huge number of professionals, we will have to share and distribute care…”(Psychiatrist-I1)

Flow management has a significant influence on the utilization of tools and raises questions about the allocation of resources in the field of psychotrauma. While many CRP professionals express the importance of responding to demand at a population level, one CRP favours an individual approach, as its professionals consider that it is not possible to outsource the care of people with C-PTSD due to the lack of resources their area. As a result, it redirects requests that have a higher prognosis for successful outcomes while retaining others. This observation underscores the interplay between flow management and the density of the healthcare network in the management of C-PTSD. The continuity of care within the network relies primarily on selected and referenced public and private partners. As mentioned earlier, this care pathway can introduce inequalities, as it is influenced by the clients’ financial means.

The treatment strategy also incorporates the concept of achieving a sufficient level of “stabilization” before initiating exposure therapy. This concept is widely shared among CRPs, but the emphasis placed on the content of this concept may vary. Some professionals believe that CRPs should primarily focus on environmental and social resources, while others prioritize addressing psychiatric comorbidities, including personality changes, or clinical aspects such as emotional regulation skills or dissociative elements. These divergences result in different evaluations and designs of the care pathway. The question of where the stabilization process takes place—whether it occurs within or outside the CRP—varies depending on the analysis and approach of each CRP. These differences lead to varying assessments and structures of the care pathway, with the concept of stabilization influencing the point of entry into care or the location of care, depending on how each CRP integrates or outsources the stabilization process within the center:

“We externalize when there is a lot of emotional regulation work before, that is to say that here, the idea is really a short, intensive care but with a limited number of sessions”(Psychiatrist-I1)

Furthermore, while some professionals hold the belief that stabilization is crucial prior to starting psychotrauma-focused treatment, others argue that initiating short-term therapy can also contribute to stabilization. They caution against excessively prolonged stabilization periods, as it may result in missed therapeutic opportunities:

“… we should not do too much stabilization if we want to stabilize client because they are already in a situation where they are destabilized, it is better to go straight to the problem, i.e., to the trauma, and so there is a period of psycho-education and empowerment training which is important, much more than saying here it is unstable or we will never do it… The objectives are really to treat the trauma, i.e., the exposure, the emotional regulation and to try to minimize the stabilization period as much as possible. Because otherwise we can spend months doing stabilization”(Psychiatrist-I9)

The increased recognition of C-PTSD within the healthcare system has acted as a catalyst for changes in the approach and strategies of psychotrauma care. CRPs have been actively involved in the development and evaluation of new care strategies such as psychocorporal care, psycho-educational groups focusing on dissociation and emotional regulation, and hotlines to facilitate referrals to health services, among others. Additionally, CRPs are exploring new perspectives to better understand the field of psychotrauma. Indeed, after an initial conception of psychotrauma care based on a “psychotraumatic emergency”, some professionals felt that optimal care for C-PTSD should be longer-term:

“… professionals are more into acute and rapid care than in long-term care which can sometimes require more complicated psychotherapeutic know-how… Often, these personalities, even in psychiatry, we no longer talk about their traumas, and we are satisfied with a semblance of balance that we try to achieve by doing this pseudo-stabilization without tackling the heart of the problem, which is security, the repetition of the trauma and so on”(Psychiatrist-I9)

Some professionals believe that it is necessary to work on personality change in order to stop the vicious cycle of constant re-traumatization; to this end, they invest in schema therapy, cognitive and behavioral therapies, or the psychodynamic approach:

“In people with C-PTSD [we should] work on lowering the triad of trauma symptoms and also accompany the person on a personality change. Because in fact the problem is that if you only work on the trauma but you don’t work on the construction of the personality, there is a good chance that the person will get back into situations that put them in danger, and as a result, they will relive the trauma and again, they will have to be taken care of. Whereas if you consolidate self-confidence or personality elements that could put them in danger, you consolidate a possible reduction in the recurrence of trauma.”(Psychologist-I13)

This view of a vicious cycle of re-traumatization complicates the ethical issues and the choice of management to strike a balance between the needs of the individual and those of the population:

“It is an ethical dilemma that consists in asking ourselves: are we offering the best possible care to very few people? Do we try to be egalitarian and give everybody the same kind of care…?”(Psychiatrist-I10)

The different results presented above explain the dynamics of care organization for C-PTSD patients. They show how the different perceptions, care strategy management, the network, and caregiver–client encounters lead to a diversity of applications of the same specifications.

### 3.3. General Model

The above results allow us to theorize a model of the different dynamics present in the construction of the care pathway of trauma survivors with C-PTSD who contact a CRP, as shown in Figure 1.

The model proposes that the degree of openness or specificity in perceiving the requirements of C-PTSD patients relies on several factors. These factors include:-The choice of theoretical framework: The theoretical framework adopted by the caregivers plays a crucial role in shaping their perceptions. Different theoretical perspectives may emphasize different aspects of the clients’ needs.-The caregivers’ relationship with diagnostic and therapeutic tools: The caregivers’ familiarity and comfort with available diagnostic and therapeutic tools also influences their perceptions. This relationship can affect the way they perceive and respond to the clients’ symptoms and needs.-The organization and content of caregiver–client interactions: The nature of exchanges between caregivers and clients significantly affects their perceptions of the clients’ needs.

By considering these factors, we can gain insights into how the openness or focus of perceptions regarding the C-PTSD clinic and its needs can be influenced. This model illustrates how, secondary to perceptions of symptoms and needs, different care strategies, the network, and the caregiver–client encounter create flexibility in the specification and diversity of roles and care pathways within different CRPs. Re-reading and exploring the model through the prism of public health policy would allow for a better understanding of the nature of the dynamics surrounding the care offered to C-PTSD clients in CRPs.

## 4. Discussion

### 4.1. Comparison with the Literature

#### 4.1.1. The Plural Essence of C-PTSD

The diversity of clinical perceptions regarding the needs and care of individuals with C-PTSD has led CRPs to use their resources (evaluation or care tools, management strategies, and the healthcare network) in different ways. The literature recognizes the variability in perceptions as a fundamental element that contributes to the universal validity of C-PTSD. In fact, the precursor diagnosis of C-PTSD in the sense of the ICD-11, the diagnosis of “disorders of extreme stress not otherwise specified” (DESNOS) [3], introduced in the appendix of the DSM-IV [4], was rejected after inconclusive field tests carried out in 2005 on post-conflict samples [24]. De Jong JT et al. [24] found that the diagnostic reproducibility of the DESNOS was undermined by its lack of homogeneity within the samples tested.The low reproducibility of the construct was attributed to the variability of symptoms in terms of type and intensity in different contexts and cultural settings. This emphasizes the substantial impact of cultural scripts, encompassing thought patterns, cognitions, emotions, and mental representations, along with social norms that dictate socially accepted ways of feeling, thinking, and behaving. These cultural factors act as confounding variables, hindering the establishment of a unified understanding of a single disorder. This variability in response to traumatic situations led the ICD-11 working group on the classification of stress-related disorders to propose the new diagnostic category of C-PTSD as a universal human response to complex trauma. The diagnostic criteria reflect the formal and structural aspects of C-PTSD according to “a process of abstraction and de-contextualization” [25]; they have been formulated in a sufficiently abstract manner to encompass the various possible facets and nuances within universally applicable dimensions. The construct of C-PTSD aims to serve as a universal framework for comprehensively capturing the complex phenomenology of mental disorders arising from severe or repeated trauma, irrespective of cultural differences. While serving as a fundamental module, this framework should be further enriched in clinical practice through the analysis of variability in dimensional expression, taking into account individual and cultural idiosyncratic components as well as potential comorbid psychotraumatic symptoms [25].

The variation in perceptions among different CRPs appears to align with the pluralistic nature of the C-PTSD diagnosis. It seems important for the CRPs to emphasize the flexible use of theoretical frameworks to effectively capture the diverse range of complex symptoms that may or may not manifest, depending on individual and cultural factors.

#### 4.1.2. C-PTSD in the Nosography

The model developed in this study proposes that the caregiver’s frame of reference profoundly modifies the clinical perception, the use of tools, and thus the care proposed by each CRP. Highlighting this dynamic within the CRPs supports the effectiveness of the WHO’s strategy in developing the C-PTSD diagnosis. In fact, by revising the PTSD diagnosis and creating C-PTSD, the WHO is introducing a new framework for reading the psychotrauma spectrum that differs from that of the DSM-5, with two main goals [26]:-To simplify the diagnosis of PTSD: the DSM-5 diagnosis of PTSD, described as 20 symptoms in 4 clusters and a dissociative subtype, results in 636,120 possible presentations, making assessment and treatment planning difficult.-To clarify the spectrum of psychotrauma and facilitate the development of personalized medicine. Specifying the spectrum by identifying greater categorical homogeneity will stimulate medical research to provide more specific and effective personalized care.

However, in the CRPs, as in the literature, the diagnostic criteria for C-PTSD, particularly the disturbances in self-organization (DSO) dimensions (emotional dysregulation, interpersonal and self-image disturbances), overlap with the symptoms of personality disorder and raise clinical questions. A recent study [27] using latent class factor analysis on a large sample observed the categorical distribution of DSO according to the use of ICD-11 or DSM-5 diagnostic criteria for PTSD and borderline disorders. The results indicate a different categorical distribution of DSO symptoms according to the reference used and confirm that DSO symptoms as described by the ICD form a homogeneous category that is distinct from PTSD and borderline disorder (as defined by the ICD). This confirms the validity results of a categorical distinction found in another study of the dimensional boundaries between PTSD, C-PTSD (PTSD + DSO), and borderline disorder [7]. Powers and others [27] also found in their study that the ICD-11 PTSD, DSO, and BPD factors showed differential relationships with external correlates (such as trauma-related avoidance, anxious attachment, and aggression), supporting their discriminant validity. The use of DSM-5 criteria on the same sample resulted in a two-category model in which the full range of DSO symptoms could also be captured and split between the PTSD and borderline personality disorder diagnoses. The authors hypothesize that the very definition of PTSD and the reduction in the number of its criteria in the ICD-11 compared with the DSM-5 leads to a reduction in the coverage of the psychotraumatism spectrum by the diagnosis of simple PTSD and allows for a more precise clinical specification of the remaining spectrum. Although the contribution of the C-PTSD diagnosis to the screening and assessment of a symptomatology can be considered weak, as this can already be captured in the DSM-5 by the association of tools for the assessment of PTSD and borderline personality disorder, the ICD-11 C-PTSD offers the possibility of refining the treatment needs by individualizing clinical profiles, whose more homogeneous dimensions are conducive to the search for specific interventions.

#### 4.1.3. CRP Targets and Strategies

Whether C-PTSD-specific or trans-nosographic DSOs are considered, the value of seeking innovation in the treatment of C-PTSD is supported by a meta-analysis that found lower efficacy of trauma-focused psychotherapies for C-PTSD after childhood trauma history [16]. Although further studies are needed to investigate the influence of DSOs on the spectrum of psychological trauma and the potential benefits of their treatment, some initiatives have shown clinical improvement in DSOs, such as mindfulness and interpersonal therapy on the interpersonal dysfunction dimension or the efficacy of dialectical behavior therapy (DBT) on the emotional dysregulation dimension [28,29]. The literature therefore suggests that the evolution of treatment toward greater effectiveness may involve a broadening of perceptions, targets of treatment, and adapted therapies, and it supports the value of understanding the dynamics between tools and perceptions modelled by outcomes. Many therapists within CRPs consider the longitudinal dimension of the development of this disorder and in particular the chronic cycle of re-traumatization. As the etiology of C-PTSD seems to be biopsychosocial, it seems that to improve the clinical picture in a sustainable way, it is also necessary to invest in the innovation of non-sanitary targets. These new targets, although offering innovative solutions, would currently lead to a considerable increase in the technical nature and duration of care, putting the principles of the obligation to provide resources in tension with the reality of the material and human resources available.

#### 4.1.4. CRPs Forced to Act as Public Health Decision Makers

Conceived and established in the aftermath of the 2015 terrorist attacks, CRPs have seen their missions and their clients evolve. They were initially assigned the role of assessing, orienting, and treating: *“All persons exposed to violence or to an event that has led or is likely to lead to psychotrauma, regardless of the anteriority of this violence:*

“-Physical, sexual, psychological violence (intimate partner violence, violence in the family, at work, in times of war, exile and migration …);

“-Exposure to a traumatic event such as an assault, accident, natural disaster, traumatic death, and others.”[30]

However, the longitudinal nature of some post-traumatic disorders has highlighted the need for adapted forms of care, especially as this population represents an important public health target. This is reported in the 2017 “Living Environment and Safety” survey [31] cited in the CRP mission statement: *“Between 2012 and 2016, an average of 364,000 people aged between 18 and 75 were victims of physical and/or sexual violence in the household each year. For its part, the National Observatory for Child Protection lists 19,700 minors who were victims of sexual violence in 2016, including 7050 victims of rape”*. These figures are in line with the current perception of chronic traumatic situations that favor C-PTSD and a need for the involvement of health, social, and legal authorities and dedicated budgets for care. However, these ambitions seem to be at odds with the resources allocated, which are insufficient to assess and provide expert care in more than ten CRPs but also to structure a network capable of absorbing the flow of needs in conditions that ensure fair access to care; these resources seem even more inadequate given that this care must also deal with the backlog of victims from previous decades. As CRPs and their networks may find it difficult to increase the volume of care provided to people diagnosed with C-PTSD, CRPs adapt and are led to deviate from their original role, and mediate care with their actual resources. In fact, the treatment strategies implemented to manage the flow of patients are restrictive in nature, whether in terms of accessibility, e.g., the use of specialist referrals; the duration of treatment at the CRP; or the aim of treatment, such as the treatment of dimensions related to personality change. As this level of specialization and accessibility cannot be guaranteed, it leads to inequalities in the care provided to those affected by C-PTSD.

### 4.2. Limitations of the Study

The novelty of the diagnosis of C-PTSD, and the fact that only half of the professionals interviewed understood it well, may have influenced the collection, understanding, and interpretation of the phenomena studied, which are bound to evolve. Furthermore, the lack of comments from the participants prior to the analysis may have hindered the anchoring of the theorized model.

### 4.3. Perspectives

#### 4.3.1. Reorganizing Care around an Emerging Disorder

The difficulty of reorganizing and transitioning care as a result of changing medical views is a common phenomenon. For Roberts et al. [32], “much of the skepticism and frustration about the scale and pace of change and innovation in the current health system stems not from a lack of vision, effort or even resources but rather from attempts to remake a model of health care that was never designed to do what is now being asked of it”. Their research team, specializing in care management and systems innovation, suggests that care actors should integrate management methods from outside the health sector into their innovation policies to steer their development effectively. They advocate for a design thinking approach to the organization of care, proposing an organizational structure that aims to strengthen the ability to recognize and articulate the explicit and latent needs and wishes of stakeholders, especially those outside the health system:-Instead of striving to identify a single optimal strategy while disregarding others, it is more productive to engage in constructive and collaborative work that embraces diverse perspectives.-Start small by rapidly testing multiple hypotheses and potential solutions in the communities most affected by persistent health problems and with the most to gain, rather than fully deploying a few purely theoretical ideas.

The design thinking innovation framework suggests testing many raw ideas in rapid iteration or prototyping, generating multiple alternative hypotheses and divergent strategies before selecting the best options available for the target communities. This framework could help answer the many questions that have not yet been adequately addressed in the literature, such as the use of stabilization practices in C-PTSD. This perspective leads to viewing variability in the organization of care, whether secondary to structural constraints or not, as an opportunity to improve care, if approached as such. The model developed here provides CRP professionals with visibility into the dynamics internal to their CRPs and to other CRPs regarding the care pathway proposed for people with C-PTSD.

#### 4.3.2. The Key Future Role of the CRPs

Our study findings highlight that effectively managing the high volume of requests poses a significant challenge in the provision of care for C-PTSD in CRPs, as well as for the professionals who receive client referrals. In order to address this issue, we propose that CRPs can play a crucial role in developing a primary prevention policy at both the local and national levels, aimed at reducing the incidence of C-PTSD in the future. By collaborating with various institutions such as healthcare, social welfare, justice, and child protection, CRPs can coordinate and implement primary prevention strategies to mitigate complex post-traumatic stress. Notably, some international researchers advocate for the implementation of primary prevention programs that target factors contributing to C-PTSD at multiple levels, including shame and humiliation. Taking these measures can contribute to a comprehensive approach to preventing the onset of C-PTSD in the long term [33]:-Relational/individual:
○Promote quality attachment and work against abuse and violence at home;○Prenatal and postnatal counseling and investment in parenting education, as a study suggests a reduction in the incidence of incest among invested parents [34].-Community:
○Social bonding tools, mobilization for investment to identify and address common problems, and a working committee to improve community life;○Recognize that individual psychological trauma care is likely to be compromised in a community context of widespread trauma and that community-level interventions may therefore be critical in creating an environment favorable to the provision of such care [35,36].-Institutional:
○Invest against bullying in schools;○Promote dignity by working to reduce the stigmatization or depersonalization of victims and to take greater account of their subjective experiences;○Intensify the identification of situations with the potential for chronic traumatization, both in primary healthcare and outside the health sector;○Raise awareness of psychological trauma in the initial training of the justice system or law enforcement;○Implement immigration management policies that respect human dignity.-Macro-social:
○Publicity campaign on psychological trauma, facilitating access to information and promoting the principles of resilience;○Reduce financial insecurity, which affects family relationships [37] and contributes to chronic traumatization through reduced social and economic mobility [38];○Reduce social and economic inequalities, which are related to a general health inequality within the population, but also to the likelihood of being exposed to adversities such as neglect, abuse, violence, discrimination, and humiliation, with multiple consequences for the child’s development, in terms of both physical and mental health [39].

## 5. Conclusions

The care pathway for individuals with C-PTSD is shaped by the perceptions held within CRPs, which are, to a large extent, driven by the utilization of theoretical frameworks and tools. This leads to a significant diversity of care strategies, encounters, and care network constructions. It seems necessary to further clarify and integrate the positioning and dynamics of the emerging concept of C-PTSD. This approach may enable CRPs to improve their performance in terms of care innovation but also in terms of screening, as the diagnosis of C-PTSD provides a fresh perspective on certain clinical presentations, notably those accompanied by personality alterations.

The constant influx of individuals chronically exposed to traumatic experiences demands, at a national level, not only the development of expertise through research and innovation, but also consideration of the goals and dynamics of care in relation to C-PTSD. Further research focusing on the subjective needs of people with C-PTSD can equip CRPs and policy makers with crucial insights for the development of relevant and effective care.

## Figures and Tables

**Figure 1 ijerph-20-06278-f001:**
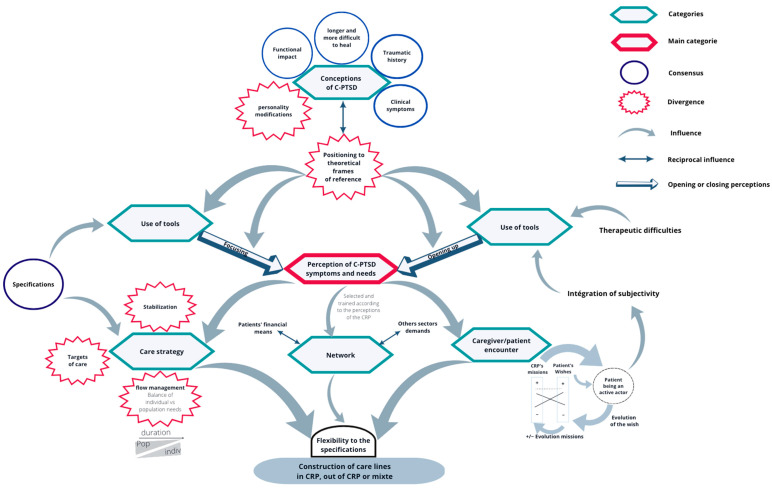
General model of the dynamics involved in building the C-PTSD care pathway.

**Table 1 ijerph-20-06278-t001:** First semi-structured interview grid-translation from French.

Question	
Main	Additional
What is your view of the role of the CRP in client care?	Does your CRP have specificities? How are clients referred to your facility? How and by whom is a client’s care plan designed (secretary, psychiatrist, psychologist, nurse, multidisciplinary consultation)?
What is your understanding of complex post-traumatic stress?	What is your definition of complex PTSD? Do you think you have a good knowledge of this new nosography? Do you think it is helpful to distinguish between C-PTSD and PTSD for treatment purposes?
How is the issue of complex post-traumatic stress disorder addressed in your CRP?	Does this clinic raise any specific questions or precautions? Is it a problem from your point of view?
How are complex PTSD clients currently managed at your CRP?	+/− Clinical client’s descriptions given to the participant: In your opinion, what should be the immediate treatment plan for this client? Over the medium and longer term? How do you imagine the current care of this client? (Which actors?) How would you ideally manage this client in your CRP?
What do you think about the role of CRP in the C-PTSD pathway?	Reception, triage, orientation, psycho-education, treatment? Do you have an opinion on care networks? What do you think about a possible standardization of practices?
Do you have anything you would like to add?	What would you like to know about how other CRPs work with C-PTSD clients? What are your thoughts on how other CRPs work with people with PTSD and C-PTSD?

**Table 2 ijerph-20-06278-t002:** Participants.

Participant	Interview Number	Length (min)	Profession	Age	Years of Experience in Trauma Care
CRP 1	I1	42	Psychiatrist	44	10
	I2	50	EMDR * nurse practitioner	44	15
	I3	53	EMDR * nurse practitioner	43	3
CRP 2	I4	68	Psychologist	53	12
	I5	53	Psychologist	39	18
	I6	82	Child psychologist	37	13
CRP 3	I7	49	Psychologist	29	4
	I8	67	Nurse	60	3
	I9	22	Psychiatrist	49	20
	I10	48	Psychiatrist	30	0.5
CRP 4	I11	62	Child psychiatrist	38	1
	I12	35	Psychologist	27	4
	I13	60	Psychologist	32	7
CRP 5	I14	74	Psychologist	46	10
	I15	29	Secretary	49	4

* Eye Movement Desensitization and Reprocessing.

## Data Availability

The data presented in this study are available on request from the corresponding author. The data are not publicly available because the whole interview transcriptions contain information that could identify the participants.

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
