# Peer review of "Determinants of Care Pathways for C-PTSD Patients in French Psychotrauma Centers: A Qualitative Study"

_ijerph, 2023, doi:10.3390/ijerph20136278_

Round 1

Reviewer 1 Report

The paper addresses an important topic and interesting methodology. But I found the writing style and the language difficult to follow. It tackles the issue of managing and characterizing complex PTSD, especially in relation to other trauma-related disorders. 

Here are some suggestions: 

- I assume the interviews were conducted in French. I found the questions in Table 1 need to be translated to standard English (e.g., “How do you see the care tasks of your CRP?”).

- It needs some proofreading (e.g., double period on page 3, line 105; “ware” on line 204).

- Individual authors contributions is not necessary in the Methods section. It should be mentioned at the end of the manuscript. 

- Please clearly define CRP. It seems to refer to different things at different places.

- Vagueness of some vocabulary choices may result in less clarity. Please consider replacing the following terminology.

1) “Dunning” in Table 1

2) “verbatim” on page 3.

3) “reflexivity” throughout the manuscript 

4) “positioning of caregivers” throughout the manuscript.

Author Response

Dear reviewer,

thank you very much for all your pertinent comments, which have enabled us to improve the manuscript considerably.

We have been able to address your comments point by point. We have also worked to make the narrative of the article more readable, as you suggested.

Please let us know if there still are specific passages in the narrative that still need reworking.

The paper addresses an important topic and interesting methodology. But I found the writing style and the language difficult to follow. It tackles the issue of managing and characterizing complex PTSD, especially in relation to other trauma-related disorders. 

Here are some suggestions: 

- I assume the interviews were conducted in French. I found the questions in Table 1 need to be translated to standard English (e.g., “How do you see the care tasks of your CRP?”). The translation of Table 1 has been revised

- It needs some proofreading (e.g., double period on page 3, line 105; “ware” on line 204). Correction of double periode /correction of the ware

- Individual authors contributions is not necessary in the Methods section. It should be mentioned at the end of the manuscript. Contributions removed

- Please clearly define CRP. It seems to refer to different things at different places. Redefined in the introduction

- Vagueness of some vocabulary choices may result in less clarity. Please consider replacing the following terminology.

1) “Dunning” in Table 1, replaced with additional

2) “verbatim” on page 3. Specified as “verbatim transcripts”

3) “reflexivity” throughout the manuscript . replaced by considerations

4) “positioning of caregivers” throughout the manuscript.   Clarification of positioning throughout the manuscript

Reviewer 2 Report

Dear authors

Many thanks for submitting this article for review and possible publication. I thoroughly enjoyed reading it and feel the readership would really engage with this given the popularity of the topic. Please find below my suggestions to strenghten the paper further. 

You have centres in the title but everywhere else in the paper its centers. Please be consistent with the spelling used. 

Although I enjoyed reading the introduction, given the topics relationship with trauma, contextualising this paper within the trauma sphere as well as the medical would be useful and of great interest to the reader. 

Line 28-30, the beginning and ending of the sentence repeats itself. 

I note how you use the word patient throughout. However, given the topic's close link to trauma informed care, can we use more inclusive language here. 

The sentence beginning in line 73 needs a reference to back up the claim. 

Line 96, you introduce DENOS without explaining what it means, please elaborate. Also, you state beside this to see appendix, however, there is no appendix to this document from what I can see. Please add. 

Your interview topic guide should be an appendix. 

You use here and throughout the paper etc, e.g and so on. Please refrain from this in academic writing and use something more appropriate such as so on. 

Line 99-111 feels choppy to read. Please reword same so that it flows better. 

Line 139, caregiver-caregiver encounter should be caregiver-client encounter. 

Please structure the document, specifically from pg 6 onwards as according to journal guidelines. All references need to be in italics and on the same font and size as the remainder of the text. 

There are also way too many quotes sitting on their own in the text as it stands. Please integrate some quotes into the text so that a clear narrative is created. This will also help with flow and allow yourselves as authors to identify the most pertinent quotes to stand out in text. 

Line 217, what is DSO? 

You refer a lot in text to borderline personality disorder. However, this is an out of date term, please use emotionally unstable personality disorder instead. 

Line 343, delete the I before In. 

Line 445, delete and keeps the others. 

Line 497, delete circle and replace with cycle. 

Line 699, newness is not appropriate here, maybe something like novelty. 

Thank you once again for submitting this paper and I look forward to reading the revision.

Author Response

Dear reviewer,

 thank you very much for all your pertinent comments, which have enabled us to improve the manuscript considerably.

We worked to make the narrative of the article more readable, as you suggested.

We have also taken on board your recommendations about quotations and worked to better contextualise them, removing those that are less useful for the reader understanding.

We have been able to address your comments point by point :

You have centres in the title but everywhere else in the paper its centers. Please be consistent with the spelling used.  corrected

Although I enjoyed reading the introduction, given the topics relationship with trauma, contextualising this paper within the trauma sphere as well as the medical would be useful and of great interest to the reader.

Line 28-30, the beginning and ending of the sentence repeats itself. corrected

I note how you use the word patient throughout. However, given the topic's close link to trauma informed care, can we use more inclusive language here., we exchange patient by other qualifications, please tell us if this sound still offensive (Individuals with C-PTSD, trauma survivors, clients, individual’s sociocultural, people who have experienced trauma event),

The sentence beginning in line 73 needs a reference to back up the claim. Reference added

Line 96, you introduce DESNOS without explaining what it means, please elaborate. Also, you state beside this to see appendix, however, there is no appendix to this document from what I can see. Please add. definition added in the introduction

Your interview topic guide should be an appendix.

We followed the journals instructions :

  • All Figures, Schemes and Tables should be inserted into the main text close to their first citation and must be numbered following their number of appearance (Figure 1, Scheme I, Figure 2, Scheme II, Table 1, etc.).

You use here and throughout the paper etc, e.g and so on. Please refrain from this in academic writing and use something more appropriate such as so on. corrected

Line 99-111 feels choppy to read. Please reword same so that it flows better. Reworded

Line 139, caregiver-caregiver encounter should be caregiver-client encounter. Corrected

Please structure the document, specifically from pg 6 onwards as according to journal guidelines. All references need to be in italics and on the same font and size as the remainder of the text. References adjusted

There are also way too many quotes sitting on their own in the text as it stands. Please integrate some quotes into the text so that a clear narrative is created. This will also help with flow and allow yourselves as authors to identify the most pertinent quotes to stand out in text. 

Line 217, what is DSO? refers to disturbance in self organisation specified in the intro

R

You refer a lot in text to borderline personality disorder. However, this is an out of date term, please use emotionally unstable personality disorder instead. This is one of the consequence due to the theorical fram finded among professionals as illustrated , a lot of professional  we met still use the DSM-5 as clinical framework, thanks to you I clarified the use of this term. Also, even if the emotionally unstable personality seems a promising way of understanding trauma, CIM-11 still reference to a borderline pattern to specify some personality disorder 6D11.5

Line 343, delete the I before In. corrected

Line 445, delete and keeps the others. 

Line 497, delete circle and replace with cycle. corrected

Line 699, newness is not appropriate here, maybe something like novelty. corrected

Reviewer 3 Report

Dear Authors, 

The paper aims at identifying the interactions of factors influencing the construction of the care pathways of the Complex Post Traumatic Stress Disorder patient through a grounded theorization method carried out with professionals working in different French structures specialized in psycho-traumatology.

In my opinion, the manuscript addresses a timely and interesting topic which might help to improve clinical knowledge and practices on care pathways of complex traumatized patients. Since the diagnosis of C-PTSD has been recently introduced within ICD-11, I believe the paper has an added value since it fills a literature gap on which clinicians need to start to work and practice.

The whole theoretical background of the paper is clear.

I would suggest to authors adding within the Introduction section some additional information about the Herman-Lewis studies and the historical development of the diagnosis of C-PTSD. Since the acceptance of this diagnosis is relatively new and the confusion of scholars and practitioners might be still high, I think that a wide and in-depth overview regarding the C-PTSD might be very useful for readers and might help to better contextualize the paper.

I have particularly appreciated that the paper uses a qualitative methodology, which I found appropriate to explore in-depth the topic investigated. Therefore, I overall consider the methodology appropriate.

Even though the results and the conclusions are consistent with the arguments presented by the authors through the manuscript and correctly address the main question posed, I think these are presented in a confused way. Firstly, the authors need to adapt into one character the paper, since I have noted different characters and size. Secondly, I understand that the figures help to “categorize” the results, however, the high number of figures make the reading too heavier and confused. The authors might try to present narratively the main results obtained as well as to summarize these into one main figures.

In general, the references are appropriate, even though the authors need to adapt the citations within the text to APA 7th edition.

Best Regards

Author Response

Dear reviewer,

 Thank you very much for all your pertinent comments, which have enabled us to improve the manuscript considerably.

We have worked to make the narrative of the article more readable and have revised the introduction as you suggested.

We have also followed your recommendations on the figures, which we hope will make the manuscripts easier to read and understand.

For the APA,7 we felt that it was really complicated to read the papers  with it and preferred to keep it in Vancouver because the IJERPH doesn't impose the type of reference.